## Research Article

translation; contextual adaptation; psychological flexibility; acceptance and commitment therapy; CompACT; Uganda

**Corresponding author:**
Joseph Mugarura;
Email: mugarurajosephh@gmail.com

# Adapting the comprehensive assessment of acceptance and commitment therapy processes (CompACT) questionnaire for contextual relevance in Uganda: A comprehensive approach

Joseph Mugarura[1] , Khamisi Musanje[2], Michael E. Levine[3], Ronald Asiimwe[4], Morris Ndeezi[1], Simon Kizito[1], Ross G. White[5] and Rosco Kasujja[1]

[1]Department of Mental Health and Community Psychology, Makerere University, Kampala, Uganda; [2]Department of Educational, Social & Organizational Psychology, Makerere University, Kampala, Uganda; [3]Psychology Department, Utah State University, Logan, UT, USA; [4]Department of Family Social Science, University of Minnesota, Twin Cities, MN, USA and [5]School of Psychology, Queen's University, Belfast, UK

## Abstract

The global utility of acceptance and commitment therapy highlights the need for adapting measures that can effectively capture the richness of psychological flexibility. One such instrument is the Comprehensive Assessment of Acceptance and Commitment Therapy Processes (CompACT). We translated the CompACT into Luganda and adapted it for use in Uganda. The original CompACT was translated into the Luganda language and reviewed through a series of evaluations. Nine mental health professionals participated in one-on-one interviews, while a focus group of eight culturally competent laypersons provided further insights. Their feedback resulted in revisions to enhance the instrument's clarity, relevance, acceptability and completeness. The revised version was then cognitively tested with $n = 25$ trainees at Makerere University. Input from these various groups was synthesized and triangulated to develop the final version. A total of 23 items were adapted to improve the comprehensibility and completeness of the scale. Overall, respondents deemed the tool clear and acceptable. This study highlights the importance of a rigorous adaptation process, including translation, expert review, cognitive testing and feedback triangulation, to ensure psychological measures remain valid and relevant across cultures. Such an approach ensures accuracy in diverse contexts and provides a model for adapting psychological instruments for non-Western populations.

## Impact statement

The Comprehensive Assessment of Acceptance and Commitment Therapy Processes (CompACT) adapted in the present study will be useful for determining the psychological flexibility levels among students and other similar populations in Uganda. Lower scores on the CompACT are consistently associated with mental health conditions and psychological suffering and, therefore, would call for support to improve mental health and psychological concerns. The findings from the process of the adaptation and validation of the CompACT may potentially serve as a guide for future adaptation and validation studies to increase survey measurements that are culturally appropriate and relevant in Uganda. It is hoped that this study will inspire other projects to adapt and validate the CompACT in other languages and cultures within Uganda. With increased contextual adaptation of the CompACT, those in need of mental health services would potentially benefit from acceptance and commitment therapy (ACT).

This study may encourage policymakers in Uganda to establish guidelines requiring the modification and validation of research measures developed outside the country before their use. Such guidelines would help ensure that the tools are culturally appropriate and generate valid, reliable data for research conducted within Uganda. The CompACT is broad in its assessment of the ACT processes/subscales, which include awareness (i.e., mindfulness of the present and flexible sense of self), openness (i.e., acceptance and cognitive defusion or flexibility) and valued action (i.e., values and committed action) and, therefore, can be useful for scholars and practitioners seeking to identify and separate the functional elements of ACT treatments that are most responsible for the change or improvement in the patient's condition.

This research is also important because it will contribute to generating data regarding the functioning of the CompACT in different contexts globally. This will enable scholars to get global estimates of the utility and functioning of the tool and also determine the efficacy and effectiveness of ACT across contexts and continents.



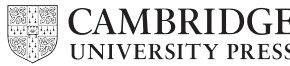

## Introduction

Mental health difficulties and disorders affect people globally in numerous adverse ways (Whiteford et al., 2013), contributing to years lived with disability, lower life expectancy and an untimely death (Charlson et al., 2015; Fekadu et al., 2015). In Uganda, it is estimated that the prevalence of mental health challenges stands at 22.9% in children and 24.2% in adults, with anxiety and depression being the commonest (Opio et al., 2022), yet diagnosis and treatment remain suboptimal across populations in many countries, including Uganda (Kohrt et al., 2018).

Improving the treatment of mental health-related conditions starts with proper assessment and diagnosis (Paniagua, 2010; Carroll et al., 2020). However, identifying and standardizing measurement tools for use in culturally diverse settings, such as Uganda, is still a challenge (Mutumba et al., 2015). The majority of tools used in Uganda to assess mental health conditions were designed and tested for Western populations and calibrated with Western norms and values, yet mental health is socially and culturally constructed (Paniagua, 2010). Uganda, an East African country of about 46 million people (Uganda Bureau of Statistics, 2024), is highly multicultural with 65 ethnic groups, nearly each group speaking a distinct language (Tulibaleka et al., 2021). English is the official language, and Luganda is the most extensively spoken language in the country (Ssentanda, 2014; Namyalo and Nakayiza, 2015). A systematic review and meta-analytic study found that depression and anxiety are the two most widespread mental health conditions in Uganda, affecting approximately one in four Ugandans (Opio et al., 2022). Other mental health conditions, such as psychotic disorders and PTSD, have also been reported among the population (Winkler et al., 2015; Mwesiga et al., 2020). Given the pervasiveness of mental conditions, culturally and contextually sensitive scales are needed that accurately measure mental health variables in Uganda.

One mental health variable that would be particularly beneficial to create culturally and contextually sensitive scales for is psychological flexibility (PF). PF refers to the capacity to be fully aware of the present moment as a conscious human and to be open to experience whatever thoughts and feelings arise while deliberately making choices that serve one's values (Hayes et al., 2006). Empirical studies indicate that high PF is associated with better mental health, better quality of life and less emotional distress and satisfaction with one's job (Gloster et al., 2021; Musanje et al., 2023; Macri and Rogge, 2024). In contrast, low PF is associated with mental health difficulties (Macri and Rogge, 2024). Moreover, PF is directly linked to a treatment approach called Acceptance and Commitment Therapy (ACT), which uses a combination of mindfulness, acceptance and value-based methods to address a wide range of mental health concerns (Hayes et al., 2006). Research has consistently found that ACT increases PF when treating various mental health concerns (e.g., depression, anxiety, obsessive-compulsive and related disorders and coping with chronic pain), and that these improvements in PF account for treatment effects (Gloster et al., 2021; Macri and Rogge, 2024). ACT has also been found to be effective in addressing several mental health concerns when adapted for use in Uganda (Gloster et al., 2020; Musanje et al., 2023). Similar to other psychological constructs, the meaning and expression of PF can vary across cultures and is, in fact, context-specific (Sabucedo, 2017; Gloster et al., 2021; Neto et al., 2024). Developing a culturally and contextually sensitive measure of PF for Uganda is crucial for advancing our understanding of mental health processes, identifying targets for treatments like ACT that aim to enhance PF, and exploring how interventions such as ACT impact PF as a key mechanism of change in culturally diverse settings (Lakin et al., 2023; Musanje et al., 2024).

Of the available PF measures to adapt to a Ugandan context, the Comprehensive Assessment of Acceptance and Commitment Therapy Processes (CompACT) (Francis et al., 2016) is particularly important as one of only a couple of measures that assess all of the components of PF, which include awareness (i.e., mindfulness of the present and flexible sense of self), openness (i.e., acceptance and cognitive defusion/flexibility) and valued action (i.e., values and committed action). Most PF-related measures assess only a subset of these PF components (Wolgast, 2014; Francis et al., 2016), which limits our understanding of how PF applies to mental health concerns and the effects of ACT components targeting mindful awareness, openness to internal experiences and valued action. The CompACT is unique in providing a relatively short (23-item) measure that includes subscales for behavioral awareness, openness to experience and valued action, in addition to a total score. The CompACT has become a widely used mental health assessment tool applied across various populations and contexts and translated into multiple languages to suit specific needs (Giovannetti et al., 2022; Chen et al., 2022).

However, while the CompACT has been used to assess various mental health conditions in Uganda, it has not been adapted for cultural salience and contextual sensitivity, despite the growing demand for culturally appropriate mental health assessment tools. Furthermore, implementing the CompACT in Uganda will not only offer an alternative, user-friendly mental health assessment tool but also facilitate better linkage to care. This study aims to adapt the CompACT for the Ugandan context, making it suitable for assessing PF in nonclinical populations. We chose a nonclinical population essentially for the initial adaptation and validation purposes, and to explore the CompACT's potential for use in a university sample.

## Methods

### Study design

This was a qualitative study aimed at translating and adapting the CompACT questionnaire for use in Uganda. The measure was translated to the widely spoken language in Uganda (Luganda), and we explored its comprehensibility (Is the translation understandable in the local language?), acceptability (Would the translated items be perceived as offensive?), relevance (Is the question or item relevant in the local culture?) and completeness (Would translating the items back into English relate to the same concepts and ideas as the original?) to the targeted context (Van Ommeren et al., 1999). This particular study is part of a larger initiative aimed at adapting and assessing the psychometric properties of the CompACT in Uganda.

### *The CompACT measure*

The CompACT is a general measure of PF. The scale has 23 items rated on a Likert scale of 0 (*strongly disagree*) to 6 (*strongly agree*), with higher scores indicating higher PF. The initial validation research (Francis et al., 2016) indicated that the scale has an overall Cronbach's $\alpha$ of .91, as well as a highly significant convergent validity ($r$ = .79) with the Acceptance and Action Questionnaire (AAQ-II) (Bond et al., 2011). Francis et al. (2016) also found that the CompACT had strong concurrent validity correlations (rs = .57–.65) with the Depression, Anxiety and Stress Scale-21

Items (Lovibond and Lovibond, 1995). Although the AAQ-II (Bond et al., 2011) is arguably the most commonly used measure of PF/ACT processes, it has been strongly criticized for measuring mainly defusion and acceptance, mostly excluding the rest of the ACT processes that are more comprehensively assessed by the CompACT (Francis et al., 2016). The total scores on the CompACT are attained by adding up responses from each subscale. Twelve items are reverse-scored before adding up the total score of the entire scale. The items to reverse score are 2, 3, 4, 6, 8, 9, 11, 12, 15, 16, 18 and 19. The full-scale CompACT total score ranges from 0 to 138, with higher scores indicating greater PF.

### Participants and sampling procedures

A total of $n = 42$ participants were involved in the study. That is, 9 local mental health professionals (MHPs), 8 laypersons and 25 students in Uganda. The MHPs were selected purposively based on their expertise in mental health care. The MHPs qualified for inclusion if they were bilingual (Luganda and English), aged 18 years and above and willing to participate. The selection of the review panel was purposively based on their expertise in mental health care, understanding of mental health assessment and comprehension of both English and Luganda languages. The qualifications, gender, age and professions of the MHPs are presented in Table 1.

A focus group (FG) of eight laypersons discussed the translation and findings from the mental health providers. The laypersons were selected purposively considering that they were proficient in their mother tongue (Luganda), aged at least 18 years and willing to participate. Table 2 shows their details.

Finally, the adapted CompACT was cognitively tested in a sample of 25 bilingual (Luganda and English) students at the School of Psychology. The students were included on the grounds that they were aged at least 18 years, bilingual (Luganda and English) and willing to participate. Further details about the cognitive test respondents are shown in Table 3.

### Adaptation procedures

We utilized a modification of the approach to prepare instruments for transcultural research proposed by Van Ommeren et al. (1999). According to Van Ommeren et al. five sequential steps need to be followed when adapting measures for varied contexts: the first step is translation of the original tool into a target language by bilingual indigenous speakers; followed by a review of the translated version by local professionals; the third step is assessment of each translated

**Table 1.** Demographic characteristics of MHPs

| Characteristics | | Number = 9 |
| --- | --- | --- |
| Age (years) | 32–40 | 3 |
| | 41–46 | 3 |
| | 47–52 | 3 |
| Sex | Male | 5 |
| | Female | 4 |
| Education level | Masters | 8 |
| | PhD | 1 |
| Employment | Psychologist | 8 |
| | Psychiatrist | 1 |
| Ethnic affiliation | Ganda | 9 |

**Table 2.** Demographics of FG participants

| Characteristics | | Number = 8 |
| --- | --- | --- |
| Age (years) | 33–40 | 3 |
| | 41–49 | 2 |
| | 50–52 | 3 |
| Sex | Male | 6 |
| | Female | 2 |
| Educational level | Primary | 6 |
| | Secondary | 1 |
| | Post-secondary | 1 |
| Ethnic affiliation | Ganda | 8 |

**Table 3.** Demographics of the students that participated in cognitive testing

| Characteristics | $N = 25$ |
| --- | --- |
| Age (years) | |
| 21–25 | 20 |
| 26–29 | 4 |
| 30–39 | 1 |
| Sex | |
| Male | 8 |
| Female | 17 |
| Education | |
| Bachelors | 19 |
| Masters | 6 |

*Note*: *n*, number of respondents.

item of the tool by nonprofessional natives in FG discussions, followed by a back-translation into the original language of the measure and, finally, pilot testing the translated version. We adjusted the steps to suit our context and resources available, as summarized in Table 4. The adjustments are discussed as follows.

In Step 1, the CompACT was translated to Luganda by a translation expert at the School of Languages, Literature and Communication of Makerere University. We chose Luganda as the target language for translation because it is the most extensively spoken language in the country (Ssentanda, 2014; Namyalo and Nakayiza, 2015).

In Step 2, the Luganda version of the CompACT was reviewed by nine bilingual MHPs. Each MHP met with the interviewer in person to review the translated measure. The interviews were audio recorded and notes were taken. The interviewer had a one-on-one discussion with each MHP to determine whether the forward translation was understandable, acceptable, relevant and complete in the local language and culture. More specifically, the researcher and MHP each had copies of the original CompACT and the forward-translated copy. The two were compared to ensure that the translation did not lose the intended meaning (completeness), was not offensive (acceptability), was culturally useful (relevance) and understandable (comprehensible). Where these four areas were questionable, improvements were suggested.

**Table 4.** Steps followed in translating and contextually adapting the CompACT in Uganda

| Step | Summary of tasks executed |
|---|---|
| 1. Forward translation by translation expert | The expert translated the original CompACT into Luganda. |
| 2. Review by mental health professionals | The Luganda translation was reviewed for its comprehensibility (i.e., Is the translation understandable in the target language?), acceptability (i.e., Would the respondent be uncomfortable honestly answering the question posed?), relevance (i.e., Is this question relevant in the target culture?) and completeness (i.e., Would translating the items back into English convey the same concepts and ideas as the original?). |
| 3. Review by laypersons | Culturally competent laypersons commented on the data provided by MHPs by clarifying whether the translation and changes suggested by professionals made sense locally. |
| 4. Cognitive testing | Changes suggested in Steps 2 and 3 were incorporated into the Luganda translation. Then, the researchers probed Psychology students about their comprehension and ease of response to the items on the adapted Luganda CompACT. |

Each interview lasted about 1 hour. Suggestions generated from the expert meetings guided the revision of the translated version.

In Step 3, the revised Luganda version of the CompACT was evaluated by eight culturally competent laypersons in an FG discussion that lasted about 2 hours. The FG discussion was audio recorded and notes were taken by a note-taker. Their deliberations were important to enrich the data by clarifying whether the translation and changes suggested by the MHPs made sense and were locally useful. The facilitator ascertained that there was consensus on the views and evaluation of each item in terms of understandability, relevance and acceptability of the translated version. Consensus was determined from open discussions in which the members exchanged views until the majority (typically six out the eight) settled on a particular position. Furthermore, the facilitator regularly summarized the key points made and sought clarification until a majority agreement was obtained. Social desirability bias was reduced by establishing rapport with the participants, exhaustively and clearly explaining the details of the study (e.g., purpose and objectives) and assuring them of confidentiality and anonymity of their identifying characteristics and/or responses.

In Step 4, the revised Luganda version was cognitively tested among students of Psychology at Makerere University, Kampala. The respondents were each provided a copy of the revised Luganda version of the CompACT and asked to write down their understanding of each of those items, state how easy or difficult it was to make sense of the items and, finally, rate each item. We then assessed whether the interpretation of the adapted items was in keeping with the original items. We also analyzed whether the recorded understanding of the modified items and their corresponding scores were reasonably comparable. In case items were considered difficult to grasp or confusing, the respondents were requested to suggest improvements. This approach of cognitive testing was based on what adaptation researchers in the country have utilized (Kasujja et al., 2022).

Students were chosen at this stage to interpret the items of the modified CompACT because the measure is chiefly intended for use among the student population. However, we believe that, when needed, the adapted tool may also be utilized in a community population, given that diverse categories of participants (translation experts, MHPs and laypersons selected from the community) were involved in the modification of the measure. On this basis, the interpretations of the adapted CompACT may be transferable to other nonclinical populations that use Luganda.

While van Ommeren's approach includes a blind back-translation step, in this study, it was not considered. We leveraged studies that suggest that back translation can fail to identify mistakes or inaccuracies in the forward translations, and the procedure is associated with inconsistent outcomes; hence, it does not necessarily yield equivalent and suitable translations (Black, 2018; Kasujja et al., 2022). Therefore, the use of mental health providers, laypersons and graduate students of Psychology as experts in the contexts gave us reasonable confidence that the review done to the forward translation was adequate. Moreover, back translation tends to emphasize the appropriateness of the original measures, which may dismiss the point of view and cultural considerations of the target population (Black, 2018), which defeats the logic behind transcultural adaptation.

### Ethical approval

The study was reviewed and approved by the Makerere University School of Health Sciences Research and Ethics Committee (MAKSHS-REC-2022-387) on December 1, 2022. Informed consent was acquired from all participants, confirmed either by signature or fingerprint in accordance with their literacy levels.

### Data analysis

Audio recordings of the interviews and the FG discussion were transcribed at Steps I and 2. We then reviewed the content of the transcripts and notes, focusing on the predetermined themes of comprehensibility, acceptability, relevance and completeness of the translations and adaptations (Kasujja et al., 2022). The changes suggested were used to come up with a revised draft version before cognitive testing. For cognitive testing, the researchers probed the trainees regarding their comprehension and ease of response to the items on the adapted CompACT in a survey where they indicated how they understood each item, rated the items, stated how easy or difficult it was to answer and whether there were any difficulties in answering. We examined whether the interpretation of the items and the corresponding scores were comparable. For items considered difficult to grasp or confusing, the respondents were requested to suggest improvements. Eventually, we combined the findings from all the participants across the different stages and triangulated them to come up with the final version of the adapted CompACT.

### Results

This section presents the results from the translation process of the CompACT, highlighting key findings from the predetermined themes (Kasujja et al., 2022) of comprehensibility, relevance, acceptability and completeness from the one-on-one interviews and the FG discussion that followed.

In Step 2, none of the MHP participants reported that any of the forward-translated CompACT items were considered shameful or

derogatory. Thus, all the forward-translated items were deemed acceptable. Four items (Items 10, 12, 20 and 21) were found to be completely understandable by all the participants. The rest (19 items) were considered to be not adequately conveying the meaning intended in Luganda. Only one item (Item 23) was completely revised because it had been mistranslated. The original Item 23 (I can keep going with something when it is important to me) was inaccurately translated in Step 1 ("*Nsobola okusigala nga ntambula nga waliwo kye sirina so nga kikulu gyendi*" that may be back translated as "*I can keep going even when I lack something important to me*"). This error was rectified following input from the MHPs and other participants. An alternative, accurate translation was proposed: "*Nsobola okusigala nga nkola ekintu bwekiba nga kya mugaso gyendi.*"

Except for Item 10, the other items were considered incomplete in some way (did not maintain the same concepts and ideas as in the source language). This was mainly because it proved difficult to integrate direct Luganda words for emotions, feelings and sensations into Items 2, 4, 11, 13, 15 and 22. So, equivalent Luganda phrases and terms were used. For instance, "*obulumi ku mutima*" was used to capture "painful emotions."

Further, "*okuwulira obubi*" was used to include "painful feelings and/or sensations." With respect to Item 4, which inquires about thoughts or feelings in a neutral manner, experts and laypersons noted that it is almost culturally irrelevant to avoid any thoughts or feelings. Rather, what is culturally meaningful is avoiding thoughts or feelings regarded as negative or distressing. Indeed, the final adaptation of Item 4 uses "*ebirowoozo ebibi oba okuwulira obubi*" for "thoughts or feelings" regarded as negative or bad. Furthermore, "emotions" were captured by "*embeera*" with examples like "*ennaku* (sadness), *ennyike* (depression) and *n'ebirala*" (and other similar emotional states). Therefore, revisions were carried out accordingly. The adaptations proposed by experts were also shared with the laypersons for discussion in Step 3.

In Step 3, a FG of 8 laypersons weighed in on both the comprehensibility and acceptability and relevance of the forward-translated items and the adaptations made in Step 2. The FG determined that 20 forward-translated items (2, 3, 5, 6, 7, 8, 9, 10, 11, 12, 13, 14, 15, 16, 17, 18, 20, 21, 22 and 23) were all understandable, relevant and acceptable. However, three items (1, 4 and 19) were judged to be less understandable and not as clear as intended in Luganda and, therefore, revisions were suggested. To improve the three items, the following changes were made.

Examples of changes include the following: first, Item 1 forward translation used the words "*ne mbikolerera*" to express pursuing important things in one's life. However, this translation essentially speaks to the physical effort made to attain something. Therefore, "ne *mbigoberera*," which means following or working toward achieving something like a dream or goal, better captures the pursuit of valued actions. Second, it was underlined that Item 4 translation missed the concept of avoiding private events considered distressing or discomforting by referring to thoughts and feelings neutrally. So, the phrase "*ebirowoozo ebibi oba okuwulira obubi*" (negative or distressing thoughts or unpleasant feelings) was endorsed as a key adaptation in Item 4. Third, it proved difficult to understand the meaning of the forward-translated Item 19. In their initial opinion, the item seemed to imply that something or someone else had taken over the individual's actions or even that one had lost his or her mind. The facilitator clarified the idea behind the item. After the clarification, the participants proposed the phrase "*mbikola bukozi*" to reflect that a person acts with limited awareness. The emphasis here is that although one is acting mechanically, it is still the same individual doing so, not someone or something else with complete control of the individual's mind.

The changes suggested by experts and laypersons were incorporated into the translated version before being cognitively tested, and these changes are indicated in Table 5. On the whole, however, the Luganda version of the CompACT was considered meaningful, relevant, acceptable and understandable.

In Step 4, the adapted Luganda version of the CompACT was cognitively tested among 25 Psychology trainees of Makerere University. In a survey, the participants indicated how easy or hard it was to interpret and respond to the items. The respondents were each handed a copy of the revised Luganda version of the CompACT and were asked to write down their understanding of each of those items, how easy or difficult it was to make sense of the items and finally rate each item. In case they encountered items considered difficult to grasp or confusing, the respondents were requested to suggest improvements. We then assessed whether the interpretation of the adapted items was consistent with ideas in the original items. In addition, we examined whether the student's understanding of the modified items and their corresponding scores was reasonably comparable.

The majority of the participants suggested that the adapted tool items were acceptable, relevant and mostly understandable. Nonetheless, minor changes were recommended to be made on nine items (4, 5, 7, 8, 13, 14, 17, 19 and 22) to make the tool friendlier in terms of the Luganda vocabulary. The respondents considered certain Luganda words (e.g., "embeera y'ememme," "Nefumitiriza," "okwanganga" and "kasoobo") complex and so simpler alternatives ("Nkola nnyo," "Nebulira" and "okuwulira bubi") were used. Overall, the adapted tool was deemed clear and relevant by the respondents. Besides the complex Luganda vocabulary that they recommended to be simplified, generally the interpretation of the items and the scores made sense, suggesting that the adaptations were understandable and yielded a measure that retains the constructs and ideas of the original CompACT. This implies that the Luganda version of the CompACT is consistent with the PF theory.

## Discussion

In this study, we describe the translation and adaptation of the CompACT for use in Uganda. Although several assessment scales have been culturally adapted in Uganda, we are not aware of any study that has adapted the CompACT. Accordingly, the present study helps to fill this gap by creating the Luganda version of the CompACT.

The process of translating and adapting the CompACT used in our study was highly participatory and ensured that the local community was also actively involved in this endeavor. Nuanced data were collected from this diversity of participants. Therefore, this people-centered approach is essential in helping the target population understand, accept and trust both the measure and the results derived from its use. Similar methods have been successfully implemented in studies conducted in other contexts (Verduin et al., 2010; Kohrt et al., 2016; Kasujja et al., 2022).

Notable adaptations were made to the Luganda version. First, single Luganda words to express feelings, emotions and sensations are nonexistent. Therefore, phrases had to be used to capture these private experiences. This adjustment was necessary to ensure the concepts were accurately conveyed in the local language.

The difficulty of translating emotions has also been documented by other translation and adaptation researchers (Goddard, 1997; Nakimuli-Mpungu, 2012). This reiterates the point that using

**Table 5.** Adaptations in English and justifications

| Item no. | Original item | Forward translation (FT) | Adaptations made by experts and laypersons (in *italics*) | Changes represented by the adaptations in English and justifications or explanations for the adaptations |
|---|---|---|---|---|
| 1. | I can identify the things that really matter to me in life and pursue them | Nsobola okuzuula ebintu ebikulu mu bulamu bwange ne mbikolerera | Nsobola okuzuula ebintu ebikulu mu bulamu bwange ne *mbigoberera* | FT: "*ne mbikolerera*" mainly speaks to physical effort made to attain something.<br>Adaptation: "ne *mbigoberera*" better captures pursing things. It means following or working toward achieving something like a dream or goal.<br>Decision: It was agreed that the pursuit of meaningful things is not limited to the use of physical effort, and thus the adaptation held. |
| 2. | One of my big goals is to be free from painful emotions | Ekimu ku biruubirirwa byange ebinene bwe butabeera na birowoozo binnumya | Ekimu ku biruubirirwa byange ebikulu bwe butabeera na *bulumi ku mutima* | FT: "*birowoozo binnumya*" means painful thoughts.<br>Adaptation: "*bulumi ku mutima*" directly translates to emotional pain in the heart. In the context of this work, it means painful emotions as felt in the heart. Culturally or linguistically, it is believed that emotions (whether considered positive or negative) reside or come from the heart.<br>Decision: Therefore, "*bulumi ku mutima*" makes more sense. |
| 3 | I rush through meaningful activities without being really attentive to them | Ebintu eby'omugaso mbikola mpapirira ne sibiwa budde bumala | Ebintu eby'omugaso mbikola mpapirira *awatali kubissako nnyo omwoyo* | FT: "*ne sibiwa budde bumala*" means not giving tasks adequate time.<br>Adaptation: *awatali kubissako nnyo omwoyo* means that not paying due attention to what one is doing.<br>Decision: It was agreed that adequate time, although necessary, is not sufficient to doing things. Instead, paying attention to what one is doing is more important. Thus, *awatali kubissako nnyo omwoyo* was preferred. |
| 4. | I try to stay busy to keep thoughts or feelings from coming | Ngezaako okubeera ne bye nkola okutangira ebirowoozo okujja | Ng'ezaako okubeera ne bye nkola okutangira *ebirowoozo ebibi oba okuwulira obubi* | FT: While the FT appears accurate, participants at different stages observed that people do not just avoid or block <u>any</u> thoughts or feelings from coming. Rather, they avoid those thoughts and feelings experienced as distressing or negative.<br>Adaptation: "*ebirowoozo ebibi oba okuwulira obubi*" were included to refer specifically to thoughts and feelings experienced as negative or distressing, thereby justifying why they are being avoided.<br>Decision: "*ebirowoozo ebibi oba okuwulira obubi*" were added to the FT for the reason mentioned above. |
| 5. | I act in ways that are consistent with how I wish to live my life | Nkola ebintu mu ngeri ekwanagana n'engeri gye njagala okubeeramu | Nkola ebintu mu ngeri *ekwatagana* n'engeri gye *nandyagadde okubeeramu mu bulamu bwange* | FT: The word "*ekwanagana*" does not clearly bring out the aspect of consistency between actions and how one wishes to live their life.<br>Adaptation: The word "*ekwatagana*" better captures consistency, thus improving the translation. Also, "*nandyagadde okubeeramu mu bulamu bwange*" reflects that there is a certain way one wishes to live their life. It is that way that is then aligned with one's actions.<br>Decision: The changes were included to improve the translation. |
| 6. | I get so caught up in my thoughts that I am unable to do the things that I most want to do | Ntwalibwa nnyo ebirowoozo ne sisobola kukola bintu bye njagala ennyo okukola | Neesanga nga ntwaliddwa nnyo ebirowoozo ne sisobola kukola bintu bye njagala ennyo okukola | FT: Fairly translated.<br>Adaptation: For grammatical reasons, it was decided that "*Neesanga nga ntwaliddwa*" is better than "Ntwalibwa."<br>Decision: "*Neesanga nga ntwaliddwa*" was adopted as a better alternative. |
| 7. | I make choices based on what is important to me, even if it is stressful | Nkola okusalawo okusinziira ku bintu ebikulu gyendi, ne bwe kiba nga kinnyigiriza | *Nsalawo* okusinziira ku bintu ebikulu gy'endi, ne bwe kibanga *kinkalubirira* | FT: Fairly translated but completeness queried.<br>Adaptation: "*Nsalawo*" was considered grammatically better than "nkola okusalawo"; also "*kinkalubirira*" was preferred over "kinnyigiriza" to capture stressful.<br>Decision: Suggested alternatives accepted and integrated into the item. |
| 8. | I tell myself that I should not have certain thoughts | Nkyeteekamu nti waliwo ebirowoozo bye sirina kubeera nabyo | *Neebulira* nti waliwo ebirowoozo bye sirina kubeera nabyo | FT: Fairly well done<br>Adaptation: The completeness of the FT was contested. "Nkyeteekamu" was questioned as the translation of "telling oneself."<br>Decision: The changes made sense and were incorporated. |
| 9. | I find it difficult to stay focused on what's happening in the present | Nkisanga nga kizibu okwekuumira ku kibaawo ekiseera ekyo | *Nzibuwalirwa nnyo* okukumira ebirowoozo *kukiba kiriwo* mu kiseera ekyo | FT: Well-translated.<br>Adaptation: Participants suggested changes to improve its grammar by using alternatives like "*Nzibuwalirwa nnyo.*"<br>Decision: Changes accepted and incorporated in the item. |

(*Continued*)

**Table 5.** (*Continued*)

| Item no. | Original item | Forward translation (FT) | Adaptations made by experts and laypersons (in *italics*) | Changes represented by the adaptations in English and justifications or explanations for the adaptations |
|---|---|---|---|---|
| 10. | I behave in line with my personal values | Neeyisa okusinziira ku ebyo bye nzikiririzaamu | Neeyisa okusinziira kw'ebyo bye nzikiririzaamu *nga nze* | FT: Fairly well-translated.<br>Adaptation: However, it was felt that the FT did not emphasize that the values are really personal. Thus, "*nga nze*" were added to the FT to underline that the individual is behaving in line with his or her personal values.<br>Decision: Changes accepted and incorporated. |
| 11. | I go out of my way to avoid situations that might bring difficult thoughts, feelings or sensations | Nfuba nnyo okwewala embeera eziyinza okundeetera ebirowoozo ebizibu | Nfuba nnyo okwewala embeera eziyinza okundeetera ebirowoozo ebizibu *oba okuwulira obubi* | FT: Incomplete feelings and sensations left out in the translation.<br>Adaptation: "*okuwulira obubi*" included to cater for feelings and sensations.<br>Decision: Accepted and integrated. |
| 12. | Even when doing the things that matter to me, I find myself doing them without paying attention | Ne bwe mba nga nkola ebintu ebikulu gye ndi, neesanga nga mbikola nga sibitaddeeko mwoyo | Ne bwe mba nga nkola ebintu ebyamakulu gyendi, neesanga mbikola sibitaddeeko mwoyo | FT: Fairly translated, but has questionable grammar.<br>Adaptation: Minor changes made to make it grammatically more accurate: e.g., used "*ebyamakulu*" instead of "*ebikulu*"<br>Decision: Changes accepted and integrated. |
| 13. | I am willing to fully experience whatever thoughts, feelings and sensations come up for me, without trying to change or defend against them | Ndi mwetegefu okufuna ebirowoozo eby'engeri yonna ebinzijira, awatali kugezaako kubikyusa oba okubiremesa okujja | Ndi mwetegefu okufuna ebirowoozo eby'engeri yonna *n'embeera zonna (ennaku, ennyike, n'ebirala)*, awatali kugezaako kubikyusa oba okubiremesa okujja | FT: Considered incomplete. Feelings and sensations not included in the translation.<br>Adaptation: *n'embeera zonna (ennaku, ennyike, n'ebirala)* were included to capture the feelings and sensations.<br>Decision: Changes endorsed. |
| 14. | I undertake things that are meaningful to me, even when I find it hard to do so | Nkola ebintu ebirina amakulu gyendi, ne bwe mba nga ndaba nga bizibu | Nkola ebintu ebirina amakulu gyendi ne bwendaba nga bizibu *okukola* | FT: Fairly well-translated. However, the FT seemed to suggest that it is the meaningful things that are hard rather than the individual finding it hard to undertake the meaningful things.<br>Adaptation: The FT was considered incomplete and "*okukola*" was added to improve the completeness of the item.<br>Decision: Accepted and incorporated. |
| 15. | I work hard to keep out upsetting feelings | Nkola nnyo okulaba nga sifuna birowoozo ebinnyiiza | *Nfuba nnyo okwetangira* ebirowoozo *ebinzijja mumbeera* | FT: Considered incomplete because the word "*Nkola*" implies working almost only in the "physical sense," yet keeping out upsetting feelings can be done in other ways. Also, "*ebinnyiiza*" speaks to "anger" rather than "upset" as in the original. Finally, "*okwetangira*" was also included to explicitly bring out "keeping out" feelings.<br>Adaptation: "*Nfuba*" and "*ebinzijja mumbeera*" were added to capture upset and thus improve the completeness of the item.<br>Decision: Accepted and incorporated. |
| 16. | I do jobs or tasks automatically, without being aware of what I'm doing | Neesanga bwesanzi nga nkola emirimu oba eby'okukola ebirala nga simanyi nti mbikola | Neesanga bwesanzi nga nkola emirimu oba eby'okukola ebirala nga simanyi nti *ndi mukubikola* | FT: It was fairly well-translated.<br>Adaptations: "*ndi mukubikola*" were added to improve the grammar.<br>Decision: The changes were accepted and integrated. |
| 17. | I am able to follow my long-term plans, including times when progress is slow | Nsobola okugoberera enteekateeka zange ez'ekiseera ekiwanvu ne mu kiseera nga bye nkola bigenda mpola | Nsobola okugoberera enteekateeka zange ez'ekiseera ekiwanvu ne mu kiseera nga bye nkola *bitambula* mpola | FT: Generally, well done.<br>Adaptation: Participants thought it that "*bitambula*" is a better word in capturing progress than "*bigenda.*"<br>Decision: Finally, it was agreed that we replace "bigenda" with "bitambula." |
| 18. | Even when something is important to me, I'll rarely do it if there is a chance it will upset me | Ekintu ne bwe kiba nga kikulu gyendi, sitera kukikola singa kiba nga kijja kunnyiiza | Ekintu ne bwe kiba nga kikulu gyendi, sitera kukikola singa kiba nga kijja kunnyiiza | FT: It was accurately translated.<br>Adaptation: No changes were suggested, as the FT was considered comprehensible, acceptable, relevant and complete.<br>Decision: The FT was maintained. |
| 19. | It seems I am "running on automatic" without much awareness of what I'm doing | Kirabika ebintu byange byekola byokka nga simanyi bulungi kiki kye nkola | Kirabika ebintu byange *mbikola bukozi* nga ssimanyi na bulungi kiki kyenkola | FT: Seemed to imply that someone or something else had taken over the individual's actions or even that one had lost his/her mind.<br>Adaptation: "*mbikola bukozi*" was instead used to reflect that a person acts with limited awareness. The emphasis is that although one is acting mechanically, it is still the same individual doing so, not someone or something else with complete control of the individual's mind.<br>Decision: The adaptation was accepted. It improved the item's relevance and completeness. |

(*Continued*)

**Table 5.** (*Continued*)

| Item no. | Original item | Forward translation (FT) | Adaptations made by experts and laypersons (in *italics*) | Changes represented by the adaptations in English and justifications or explanations for the adaptations |
|---|---|---|---|---|
| 20. | Thoughts are just thoughts – they don't control what I do | Ebirowoozo biba birowoozo bulowoozo – <u>tebisalawo</u> kiki kye nkola | Ebirowoozo biba birowoozo bulowoozo – *tebifuga* kiki kye nkola | FT: The word "tebisalawo" is more about decision-making rather than control as used in the original item.<br>Adaptation: Since the FT was considered incomplete, the word "*tebifuga*" was adopted as better reflecting control.<br>Decision: It was agreed that "*tebifuga*" improves completeness and thus the preferred alternative. |
| 21. | My values are really reflected in my behavior | Bye nzikiririzaamu byeyolekera mu nneeyisa yange | Bye nzikiririzaamu by'eyolekera *ddala* mu nneeyisa yange | FT: The translation was almost perfect but left out the word "really" and thus made the item incomplete.<br>Adaptation: "*ddala*" was added to capture the "really" in the original item, thereby improve completeness.<br>Decision: The adaptation was accepted and incorporated. |
| 22. | I can take thoughts and feelings as they come, without attempting to control or avoid them | Nsobola okutwala ebirowoozo nga bwe bizze, awatali kugezaako kubirwanyisa oba okubyewala | Nsobola okutwala ebirowoozo *n'embeera (ennaku, ennyike, n'ebirala)* nga bwe bizze, awatali kugezaako *okubifuga* oba okubyewala | FT: It only captured thoughts (*ebirowoozo*) leaving out feelings.<br>Adaptation: Feelings were included by using the word "*embeera*," which can be translated as an "internal state." Examples of the states specifically referred to were captured by extra words: *ennaku* (*sadness*), *ennyike* (*depression*) and *n'ebirala* (and other similar states).<br>Note: There is no one Luganda word for "feelings," which is why we adopted the "*embeera*" phrase followed by examples.<br>Decision: The changes were accepted and integrated. |
| 23. | I can keep going with something when it's important to me | Nsobola okusigala nga ntambula nga waliwo kye sirina so nga kikulu gyendi | Nsobola okusigala nga *nkola ekintu bwekiba nga kya mugaso gyendi* | FT: The original item was inaccurately translated. The forward translation read like: *I can keep going with something even when I don't have something that's important to me.* The domain of completeness was not met.<br>Adaptation: To make the translation complete, we adopted this: "Nsobola okusigala nga *nkola ekintu bwekiba nga kya mugaso gyendi.*"<br>This better brings out the point that one can carry on/persist with something as long as it is important to them.<br>Decision: The adaptation held. |

questionnaires without adapting and validating them for the target contexts and populations can yield incomplete or erroneous data (Epstein et al., 2015).

The second adaptation (especially on Item 4 of the Luganda version) pertained to clarifying that, culturally, only private events (feelings, emotions, thoughts and sensations) considered unpleasant or distressing are avoided or resisted. This adaptation is in line with the assertions of ACT theory, which asserts that people tend to avoid the private events they experience as painful/negative/unpleasant. ACT calls this psychopathological process experiential avoidance (Harris, 2006; Hayes et al., 2006).

Another notable adaptation that had to be made for concerned item number 19. According to the FG participants, the phrase "running on automatic" had been forward translated in such a way that it could be mistaken to imply that something or someone else had taken over one's actions. Alternatively, a few thought it meant "losing one's mind." The facilitator helped clarify this misinterpretation and better phrasing suggested by the participants improved comprehension. Therefore, conceptual coherence was eventually achieved.

The translation and modifications of the CompACT still maintained the number of items and subscales as in the original measure. This shows that even though the process of modifying the measure was lengthy and rigorous, the changes replicated a version that is consistent with the PF, albeit in a different cultural context. This suggests a theoretical coherence across the source and target cultures, implying that PF may transcend cultures (Lin et al., 2020; Giovannetti et al., 2022).

### *Strengths and limitations*

Research indicates that adaptation and validation studies can involve methodological weaknesses (Beaton et al., 2000; Borsa et al., 2012; Haroz et al., 2014; Yang et al., 2022). For instance, it was found that many scholars simply do forward and back-translation only and then claim they have adequately adapted survey measures/tools (Beaton et al., 2000). Other scholars reveal that adaptation and validation studies do not adequately outline clear procedures for adapting scales (Borsa et al., 2012; Yang et al., 2022). To obviate the above flaws, a modification of the van Ommeren et al., approach was utilized in this study. Compared to the simple forward and back translation, the approach we used is more rigorous and costlier in terms of time, money and human expertise, but it increases chances to more thoroughly iron out any errors, weaknesses or inconsistencies that may be missed by mere forward and back translation. This contributes to producing better measures. Other studies in Africa and beyond have used the same approach with success (Verduin et al., 2010; Kohrt et al., 2016).

However, the findings from the present study are not generalizable to the rest of the population in the country because the study was limited to experts and students in Makerere University and laypersons who were fluent in Luganda. Notably, the psychometric properties (factor structure, reliability and validity) of the adapted CompACT are unknown; therefore, future studies should investigate its psychometric properties. In addition, Luganda is just one of the numerous languages in the country. Thus, future research should culturally modify this tool in other Ugandan languages.

### Implications and conclusion

This study may encourage policymakers in Uganda to establish guidelines requiring the modification and validation of research measures developed outside the country before their use. Such guidelines would help ensure that the tools are culturally appropriate and generate valid, reliable data for research conducted within Uganda. The CompACT questionnaire is broad in its assessment of the ACT processes/subscales, which include awareness (i.e., mindfulness of the present and flexible sense of self), openness (i.e., acceptance and cognitive defusion/flexibility) and valued action (i.e., values and committed action) and, therefore, can be useful for scholars and practitioners seeking to identify and separate the functional elements of ACT treatments that are most responsible for the change or improvement in patient's condition.

Further, the findings from the process of the adaptation and validation of the measure may potentially serve as a guide for future adaptation and validation studies to increase survey measurements that are culturally appropriate and relevant in Uganda. For instance, future projects may find the Van Ommeren model useful for adaptation studies.

It is remarkable that our adaptation efforts suggest that, as measured by the CompACT, PF is consistent across the source and target cultures. Given this consistency of PF across the cultures, people of different cultures can benefit from ACT, especially if the intervention is appropriately adapted to their culture and context. This calls for additional efforts to adapt PF measures and ACT across cultures with the aim of increasing PF and its associated benefits (Gloster et al., 2021; Macri and Rogge, 2024).

Finally, the CompACT adapted in the present study may also be modified for use in other populations and cultural contexts in Uganda. With increased contextual adaptation of the CompACT, those in need of mental health services would potentially benefit from ACT.

Conclusively, results suggest that the CompACT was successfully adapted for use in Uganda. Indeed, the findings suggest that the adapted CompACT is comprehensible, relevant and appropriate to measure PF.

**Open peer review.** To view the open peer review materials for this article, please visit http://doi.org/10.1017/gmh.2025.10018.

**Supplementary material.** The supplementary material for this article can be found at http://doi.org/10.1017/gmh.2025.10018.

**Data availability statement.** The datasets generated or analyzed during the current study are available from the corresponding author on reasonable request.

**Author contribution.** Conceptualization: Joseph Mugarura, Rosco Kasujja and Khamisi Musanje. Data curation: Joseph Mugarura. Formal analysis: Joseph Mugarura, Rosco Kasujja, Khamisi Musanje, Kizito Simon and Morris Ndeezi. Funding acquisition: Not applicable (the study is not funded). Investigation: Joseph Mugarura. Methodology: Joseph Mugarura, Rosco Kasujja and Khamisi Musanje. Project administration: Joseph Mugarura. Resources: Joseph Mugarura and Rosco Kasujja. Supervision: Rosco Kasujja, Khamisi Musanje and Simon Kizito. Validation: Rosco Kasujja and Khamisi Musanje. Visualization: Joseph Mugarura, Khamisi Musanje and Ronald Asiimwe. Writing – original draft: Joseph Mugarura (lead), Khamisi Musanje and Rosco Kasujja. Writing – review and editing: All the authors.

**Financial support.** This research received no specific grant from any funding agency, commercial or not-for-profit sectors.

**Competing interests.** The authors declare no potential conflicts of interest with respect to the research, authorship and publication of this article.

**Ethical approval.** This study was approved by the Makerere University School of Health Sciences Research Ethics Committee (approval number MAKSHS-REC-2022-387) on December 1, 2022.

**Informed consent statement.** Informed consent was acquired from all participants, confirmed either by signature or fingerprint in accordance with their literacy levels.

The Luganda version of the CompACT is available in the supplementary material.

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
