## [Reviewer Report]

This study adapted a measure of psychological flexibility, which is often used to assess mechanisms of effectiveness of Acceptance and Commitment Therapy, for use in Luganda.

I appreciate the level of detail provided about the methods of cultural adaptation. My main feedback is that I’d like to see more detail about the specific adaptations that were needed, as well as consideration of implications for cross-cultural adaptation studies more broadly. CompACT is a somewhat unique measure compared to typical mental health assessment tools, so it provides an interesting example to examine challenges of translating and adapting particularly abstract or complex concepts. The manuscript doesn’t provide much detail about the actual adaptations made, and the discussion was very short. Expanding in these two areas would help strengthen this manuscript’s contribution to the literature.

Is it possible to add a column in table 5 describing in English the type of changes represented by the adaptations indicated? Even better would be including justifications or explanations as well. If not, it would be good to add more specifics about adaptations in-text since there are currently only a couple examples of the changes made, mostly in step 2 and one example in step 4. It might also help to list the original English item as a column.

Could you clarify what you mean by creating terms for feelings, terms, and sensations that didn’t have equivalent translations (pg 11)? Is it that phrases had to be used rather than single terms?

To what extent does recruiting graduate student participants from a School of Psychology potentially bias the sample compared to a community-based sample? In other words, would their interpretations of items be transferable to the broader population? Alternatively, is the student population the target population for future use of the CompACT (in which case that should be stated)?

I wasn’t entirely clear how cognitive interview data were collected. Was it a survey completed with an enumerator?

Consider replacing the word “natives” with another word.

It seems likely that the information provided about the initial translator is specific enough to be identifiable. It would be good to clarify that they’re ok being identified in this way.

---

## [Reviewer Report]

Thank you for asking me to review this study, which outlines the translation of the Comprehensive Assessment of Acceptance and Commitment Therapy Processes (CompACT) into Lugandan and the Ugandan cultural context.

The paper makes a good case for the necessity of this translation, highlighting cultural differences and the fact that the CompACT and similar measures of psychological flexibility have largely been developed and validated within Western cultures and norms. The authors argue convincingly that measures should be culturally sensitive to retain contextual meaning, rather than being simply directly translated and back-translated.

Given that Uganda has similar levels of mental health needs, the authors outline the need for appropriate measures that can be used as therapy outcome measures. I agree with the authors that such translational work is essential to improve mental health across different contexts.

The work itself is commendable and methodologically well-grounded, drawing on and adapting established protocols for cultural measure translation. The use of an expert translator and the multiple steps of checking and validation are significant strengths. In my opinion, the paper will make a useful contribution to the literature and ACT practice within Uganda.

There are, however, some areas that would benefit from further attention:

• The authors highlight that they aim to test the Comprehensibility, Acceptability, Relevance, and Completeness of the translations. This is essential, but it would be useful to operationally define each of these domains, and highlight how they were assessed. This is particularly important for relevance; in addition to cultural relevance, it is arguably crucial that the translated items still map onto the same theoretical constructs they were designed to measure. While the items may be acceptable, it would be useful for the authors to describe how the theoretical coherence of the items was maintained during the translation and feedback process. Highlighting how conversations at each step helped shape the items while maintaining theoretical consistency (or noting where this was not the case) would be particularly useful. Essentially, it is important to know whether the translated items still adhere to psychological flexibility theory, in addition to making linguistic sense and adhering to cultural norms.

• The cognitive testing element would benefit from further elaboration. Was this based on a particular cognitive interviewing framework? How was understanding assessed? Was there any assessment of whether participants' verbal understanding mapped to the response options they chose? If this element was not examined, that is defendable, but more information on how understanding was checked—and importantly, whether understanding continued to map to psychological flexibility theory—would strengthen this section.

• The discussion section summarises the work well. However, it would be useful to discuss the translation work specifically in relation to psychological flexibility theory more thoroughly. While this is done to some extent (e.g., when outlining the difficulty in translating emotions), it would be beneficial for the authors to convey how psychological flexibility is conceptualised/understood differently when ported to a different cultural context (if indeed the conceptualisation does change), and how this compares and contrasts with the original version of the construct. This would add a unique theoretical contribution to the literature.

Smaller issues include:

• P3 L34: I’m not sure that PF being related to ACT is more important.

• P4 L68: Up to this point, most of the rationale for the measure has mentioned clinical populations, but then a non-clinical population is introduced/utilised, which reads as a bit jarring. Perhaps edit to indicate this is just for initial validation purposes.

• P5 L82: Missing ‘e’ in “The”.

• P5 L84: Given the ubiquity of the AAQ, it would be useful to outline that this is the measure most frequently used as a measure of PF, but has limitations (and references) for that debate.

• P5 L85: Unclear which correlations (or with what measures) are being discussed here.

• P6 L109: Only need the reference once here.

• P7 L133: How was consensus determined here?

• It would be useful to include the original English item next to the translated items in Table 5.

Overall, this paper presents a worthwhile study that can make a useful contribution to the literature, and ACT research and practice, within Uganda. I recommend accepting the paper with some revisions to address the points mentioned above.

---

## [Editor Report]

May you kindly address the suggested methodological and academic writing issues pointed out by the reviewers.

---

## [Reviewer Report]

I appreciate the detailed responses and edits, which have strengthened the clarity of the manuscript. I especially appreciate the level of detail and transparency throughout the manuscript about how the translation process was conducted and how decisions were made, which should serve other researchers well as they conduct similar studies. The additional column in the table that details adaptations and justifications is particularly useful. Could this table be included in the main text rather than treated as a supplemental table?

I have a few very minor suggestions:

Note that on top of pg 6, “natives” is still used

On the bottom of pg 8, I find it confusing that the wording is “the researchers probed the trainees,” as it implies use of an enumerator to collect responses. I think just saying something like a survey elicited their comprehension would be clearer.

On pg 11, where you mention “forward translated item 19,” could you briefly state the item (as you do in the discussion)?

---

## [Editor Report]

Congratulations on the provisional acceptance of your manuscript. Kindly address the minor comments suggested by the second reviewer.